# Study on Bulge State of PCrNi3MoV Target Plates Penetrated by Square Fragments

Dou Hong [1], Wenbin Li [1,*], Yu Zheng [1], Ning Jiang [1], Kebin Zhang [1], Zhonghua Liu [2] and Yao Jiang [2,*]

1 Ministerial Key Laboratory of ZNDY, Nanjing University of Science and Technology, Nanjing 210094, China; hongdounjust@163.com (D.H.); zhengyu@njust.edu.cn (Y.Z.); njustjn@163.com (N.J.); kb2018@njust.edu.cn (K.Z.)
2 School of Materials Science and Engineering, Nanjing University of Science and Technology, Nanjing 210094, China; zhonghualiu@163.com
* Correspondence: lwb2000cn@njust.edu.cn (W.L.); jiangyao@njust.edu.cn (Y.J.)

**Abstract:** The objective of this study was to study the penetration characteristics of tungsten alloy fragments on PCrNi3MoV target plates. Specifically, the penetration of square tungsten alloy fragments of gun steel target plates of different thicknesses were studied using a ballistic gun launching device. Based on the ballistic test results, the relationship between the bulge height of the target plate and the kinetic energy of the fragments was obtained, and the failure mode and process of the tungsten alloy fragments penetrating the steel target were analyzed. Then, the variation on the bulge height of the target plate with the kinetic energy of the fragments was obtained using dimensional analysis. The prediction results were consistent with the test results, and thus, the prediction model can be used in practical applications.

**Keywords:** square fragments; PCrNi3MoV target plates; bulge height





## 1. Introduction

With the development of the weaponry industry, the damaging power of ammunition has become a focus of research for the weaponry and protection industries. For example, in order to maximize the power of ammunition on the battlefield, the loading of tungsten-prefabricated fragments in projectiles has become popular in modern arms.

Tungsten alloys have high densities, strengths, and hardness values, as well as good armor-piercing abilities. Therefore, the study of the penetration characteristics of tungsten fragments is of great value. Hafizoglu et al. [1] studied the effect of the sintering temperature and Ni/Fe ratio on the ballistic performance of tungsten heavy alloy fragments penetrating Al 6061-T6 target plates. Zhu et al. [2] conducted theoretical analysis of the adiabatic shear sensitivity of tungsten alloys based on a material constitutive model. Specifically, for cylindrical fragments penetrating a target, the penetration process and velocity characteristics of steel target plates penetrated by tungsten alloy fragments were analyzed using the finite element software ANSYS/LS-DYNA and a Lagrangian algorithm. The effect of fragment shape on penetration results was also investigated. Dey et al. [3] studied the influence of the fracture criteria in numerical simulations of the perforation process of high-strength steel plates and evaluated the qualitative and quantitative differences between the Johnson–Cook fracture criterion and the Cockcroft–Latham fracture criterion. The penetration characteristics of tungsten alloy fragments were mainly studied from the perspectives of the constitutive model and the material ratio.

To evaluate the penetration performances of tungsten alloys on target plates, previous studies primarily focused on the ultimate penetration velocity and the penetration depth. Huang et al. [4] established a model of the ultimate penetration velocity of spherical fragments by combining dimensional analysis and multiple linear regression; this model had high calculation accuracy. Zhang et al. [5] established a theoretical model of the

ultimate penetration velocity of fragments from an energy perspective and used reliability theory to determine the testing method for the ultimate penetration velocity for the first time. Zhu et al. [6] calculated the penetration resistance based on cavity expansion theory and decomposed the penetration process into A and B models, and the penetration equation of each model was presented. Cheng et al. [7] studied the critical bounce and penetration velocities of a tungsten alloy rod obliquely penetrating a finite-thickness metal plate. The results showed that the test results were in good agreement with the calculation and numerical results of the critical penetration velocity model proposed by Zhao et al. [8]. However, the calculation results in the model established by Recht et al. [9] were slightly smaller than the test results. In the study of the fragment penetration in a finite-thickness target plate, the ultimate penetration velocity is usually used to characterize the penetration ability of fragments into the target plate. In 2021, Hong et al. [10] used the dimensional analysis method to study the relationship between the target penetration energy required by the fragment and the fragment velocity when the tungsten alloy fragment completely penetrates the carbon fiber target plate, and established the corresponding prediction formula. When studying the penetration problem of an infinite-thickness plate or when the fragments are not able to penetrate through the target plate, the penetration depth is used to characterize the penetration performance. In 2001, Schafer et al. [11] proposed a theoretical model for predicting the penetration depth of a rotationally symmetric ellipsoid tungsten projectile impacting a semi-infinite target for hyper velocity. Then, the shape effect of the projectile on the penetration depth was further analyzed via numerical simulations. Martineau et al. [12] investigated the penetration characteristics of high-strength low-alloy (HSLA-100) steel plates when impacted by tungsten carbide spheres at striking velocities between 0.8 and 2.5 km/s. The results showed that the diameter of the crater increased linearly with the striking velocity. However, there was a nonlinear relationship between the penetration depth and the striking velocity. In 2011, Xu et al. [13] studied the fragments after piercing armor and presented an empirical equation for the relationship between the diameter of the fragment and the ultimate penetration depth.

In terms of the ballistic performances of fragments on a target, in addition to the penetration depth and ultimate penetration speed, the bulge height of the target plate is also an important indicator for evaluating the damage. When the barrel is penetrated, the bulge height, rather than the ultimate penetration velocity, is very important for evaluating the degree of damage to the barrel. Although there are currently numerous studies on the ultimate penetration velocity and penetration depth, few studies have been carried out on the bulge height. Xu et al. [14] studied the penetration resistances of target plates with the same thickness and different number of layers by analyzing the influence of the plate thickness and the number of layers on the bulge height of the target plate. Mo et al. [15] investigated the radius and bulge height on ceramic-ultra-high molecular weight polyethylene (UHMWPE) composite target plates. The equations of motion of the projectile penetrating a ceramic cone and the ceramic cone penetrating a back plate were established, and the accuracy of the equations was verified by tests.

To the best of our knowledge, there are no studies on the penetration of tungsten alloy fragments into PCrNi3MoV gun steel. Therefore, in this study, we carried out fragment penetration tests to investigate the failure characteristics of a gun steel plate impacted by tungsten alloy fragments. Based on dimensional analysis, the relationship between the bulge height of the steel plate and the velocity of the fragments was obtained. A 14.5-mm ballistic gun was then used to launch tungsten alloy fragments for armor-piercing tests, and the test results were in good agreement with the calculated values.

## 2. Fragment Penetration Test

Aiming at the problem of prefabricated fragments penetrating flat target plates made of PCrNi3MoV gun steel, square prefabricated tungsten alloy fragments used in projectiles were used in this study to study the penetration performances of prefabricated fragments at different muzzle velocities. The composition of the gun steel is shown in Table 1.

**Table 1.** Chemical composition of PCrNi3MoV (%).

| C | Si | Mn | Cr | P | S | Mo | Ni | V |
|---|----|----|----|----|----|----|----|----|
| 0.39 | 0.32 | 0.5 | 1.10 | 0.025 | 0.020 | 0.28 | 1.80 | 0.1 |

The tests were carried out using a 14.5-mm ballistic gun. The layout of the test is shown in Figure 1. The equipment used in the test included a 14.5-mm ballistic gun, gun mount, guard plate, target plate, target frame, and high-speed camera.

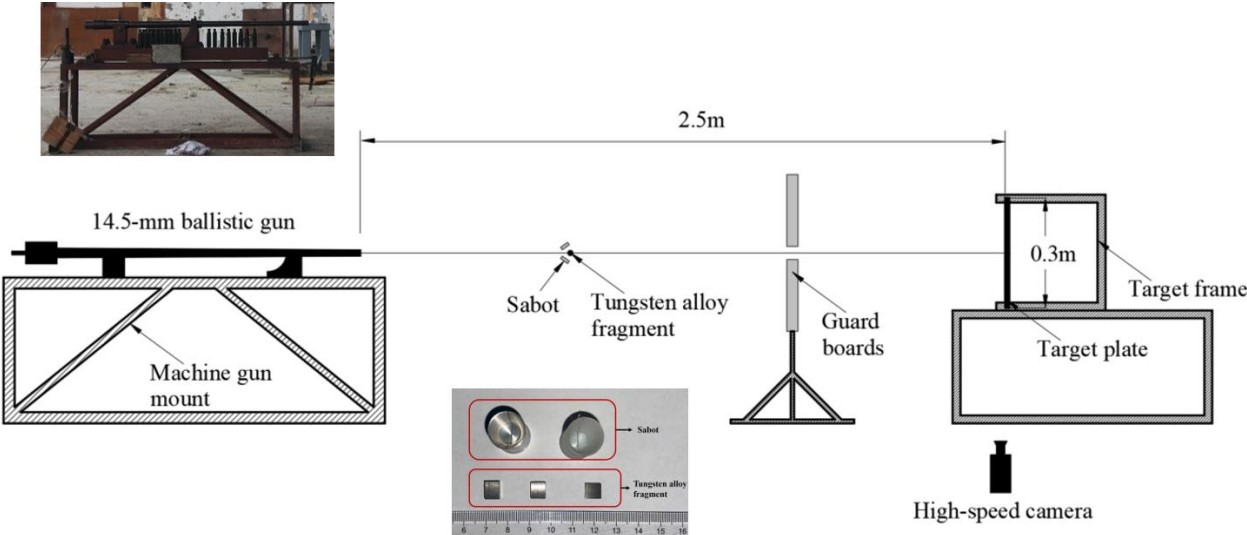

**Figure 1.** Setup of the ballistic impact test.

The tungsten alloy fragments were first put into the sabot, and the muzzle velocities of the fragments were adjusted by the amount of propellant. The maximum fragment velocity could reach 1000 m/s. After the fragments exited the muzzle, they were separated from the sabot to impact the target plate through a hole in the protective plate, while the sabot was intercepted by the protective plate. The penetration characteristics of the tungsten alloy fragments on PCrNi3MoV steel target plates with different thicknesses in the normal direction were studied. The main parameters of tungsten spherical fragments and the steel target are shown in Table 2.

**Table 2.** Main parameters of the fragment and the target plate.

| Tungsten Alloy | | PCrNi3MoV Steel | |
|---|---|---|---|
| **Density (g/cm$^3$)** | **Yield Strength (MPa)** | **Density (g/cm$^3$)** | **Yield strength (MPa)** |
| 17.8 | 1500 | 7.80 | 780 |

Ballistic performance tests of gun steel targets with different thicknesses were then carried out using square fragments with side lengths of 6-mm. In order to measure the bulge height of the target plate, the penetrated parts of the target were cut, and the axial displacement of the center point of the impact surface of the target plate, i.e., the maximum deformation was used as the bulge height $h_g$, as shown in Figure 2. When the target plate was not perforated, the larger the bulge height was, the greater the damage was.

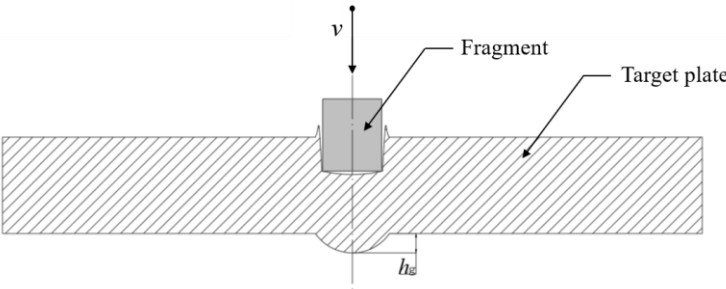

**Figure 2.** Schematic diagram of the fragment penetrating the target plate.

The impact damage modes of the gun steel target plate were studied, which included cratering, compression, shearing, and tension. Figure 3 shows the 6-mm target plate impacted by the fragment at different striking velocities. From the top view, when the fragment velocity was 277 m/s, the penetration depth was shallow, and the fragment left a small square pit on target plate. Moreover, due to the reverse force of the target plate, the fragment bounced back. As the fragment velocity increased, the degree of damage to the target plate gradually increased, as indicated by the size and depth of the pit in the frontal view and the bulge height in the side view. When the muzzle velocity of the fragment was less than the ultimate penetration velocity and the fragment embedded in the target plate to a certain depth, it left a rectangular hole in the plate, and the higher the velocity of the fragment, the deeper the hole. From the side view of the cut target plate, as the velocity of the fragment increased, the bulge height of the target plate increased. When the fragment velocity reached 658 m/s, the target plate was sheared and blocked, at which point, the ultimate penetration velocity was reached. The comparison of the pit depth and the bulge height showed that the pit depth was larger than the bulge height, and the greater the penetration kinetic energy, the larger the difference. This was because, in addition to tensile failure, the target plate was also subjected to compressive failure. Tensile failure caused a bulging phenomenon on the back of the target plate, and the compressive failure caused the pit depth to be larger than the bulge height.

To illustrate the impact damage modes, the microstructure of the gun steel target plate impacted by the fragment at different striking velocities was characterized by using scanning electron microscopy (SEM). All characterized planes were conducted on the longitudinal sections of the gun steel target plate, i.e., observing from the top view as shown in Figure 3. Before SEM observations, the samples were carefully ground and polished first mechanically to a mirror-like finish, and finally etched by 4% $HNO_3$ + ethanol solution.

Figure 4 shows the macro-morphology and microstructure of the gun steel target plate after impact by the tungsten fragment at different striking velocity. The fragment with a velocity of 277 m/s resulted in a bulge height of only 0.26 mm, thus the microstructure deformation was almost negligible. It can be identified by the identical microstructure to the matrix, consisted of randomly oriented lamellar pearlites and equiaxed ferrites (Figure 4(a1)). It is necessary to note that the initial steel was fabricated by rolling and subsequent thermal-treatment, leading to the initial fibers were parallel to the rolling plane and perpendicular to the penetration direction. With the increasing of fragment velocity up to >406 m/s, obvious penetration hole can be observed in their contact regions as shown in Figure 4b,c. The initial horizontal fibers were seriously distorted and compressed to align to the contact surface, as sketched by Figure 4d. Besides, the pile-up formation around the penetration hole is pronounced, as the plastic zone is confined and the material deforms to the free surface [16]. The target steel impacted by a fragment with a velocity of 597 m/s, as shown in Figure 4c, showed almost identical microstructure characters to Figure 4b, except some micro-cracks were identified around the contact region and marked by red arrows. It can be reasonably understood that these cracks resulted from the points of the high local plastic strain and high local stress concentration due to the high penetration speed, since

the material flow to the free surface was suppressed and the materials were compressed to overcome the constraint effect from the plate thickness.

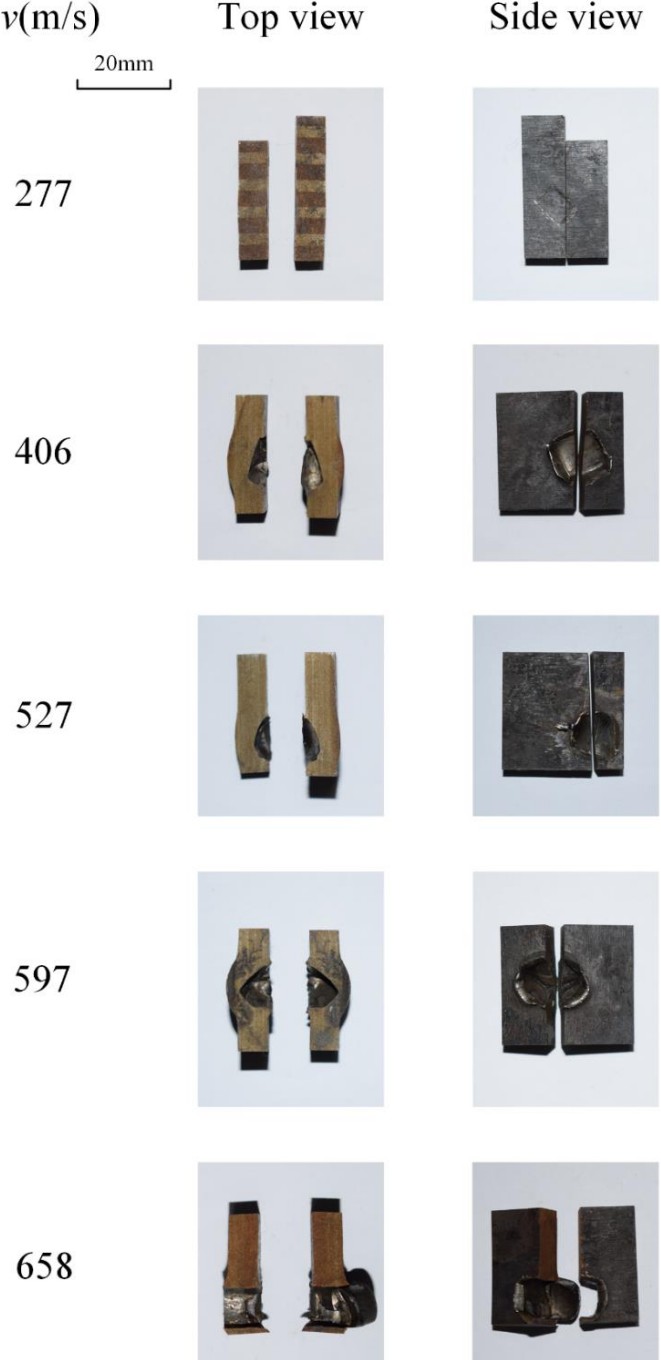

**Figure 3.** Failure modes of the target plate.

The local microstructures as marked by red squares were investigated in detail at high magnifications and given in Figure 4(a1,b1,b11,b2,b21). Figure 4(b1,b2) showed that the initial random orientated pearlite were compressed to align to the contact surface, consistent with that the horizontal macro-fibers were deformed and adjusted to the same direction as shown Figure 4b–d. Higher magnification for the contact surface region in Figure 4(b11,b21) demonstrated that the deformation introduced by penetration was heterogeneous and gradient. The deformation at the surface region was the most severe and the initial lamellar pearlites were even fragmented to equiaxed nano-grains of about

400 nm. The thickness of the nano-grains layer was measured to be about 10~20 μm, while the thickness of compression affected layer was about 500 μm.

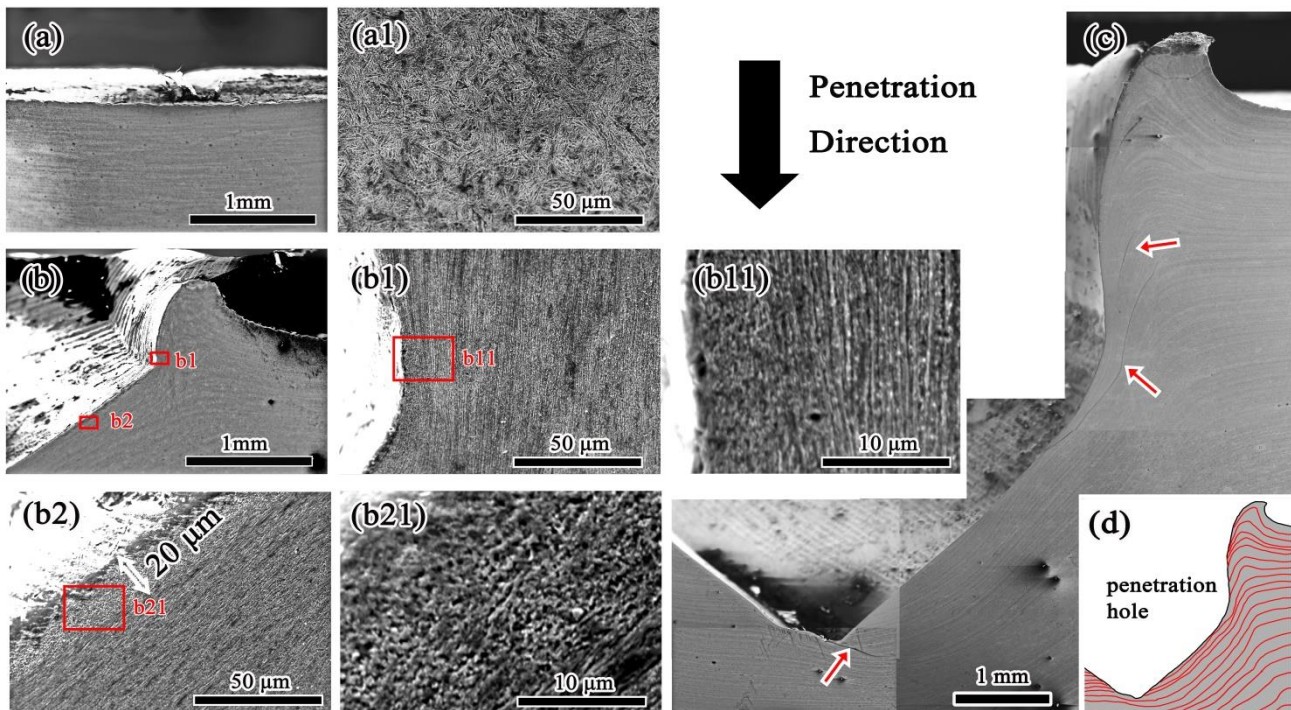

**Figure 4.** SEM images of the gun steel target plates impacted by the fragment: Morphology of the contacted region for the gun steel target plate impacted by the fragment with a velocity of (**a**) 277 m/s, (**b**) 406 m/s and (**c**) 597 m/s, respectively. The local microstructures as marked by red squares are observed in detail at high magnifications and given in (**a1,b1,b11,b2,b21**). Typically, the inset sketch map (**d**) indicates the orientation distribution for the fibers plastic flow introduced by penetration in (**c**). The red arrows indicate micro-cracks, while the black arrow indicates the penetration direction.

A high-precision altimeter was used to measure the bulge height of the target plate. The resolution of the altimeter was 0.01 mm, which was suitable for this study. The results are shown in Table 3.

**Table 3.** Bulge height of the gun steel target plate.

| No. | Fragment Length (mm) | Target Plate Thickness (mm) | Striking Velocity (m/s) | Striking Kinetic Energy (J) | Bulge Height (mm) |
|---|---|---|---|---|---|
| 1-1 | 6 | 6 | 277 | 153 | 0.26 |
| 1-2 | 6 | 6 | 370 | 274 | 0.56 |
| 1-3 | 6 | 6 | 406 | 330 | 0.8 |
| 1-4 | 6 | 6 | 452 | 410 | 1.23 |
| 1-5 | 6 | 6 | 455 | 414 | 1.3 |
| 1-6 | 6 | 6 | 455 | 414 | 1.52 |
| 1-7 | 6 | 6 | 468 | 438 | 1.3 |
| 1-8 | 6 | 6 | 503 | 506 | 1.47 |
| 1-9 | 6 | 6 | 527 | 556 | 1.62 |

**Table 3.** *Cont.*

| No. | Fragment Length (mm) | Target Plate Thickness (mm) | Striking Velocity (m/s) | Striking Kinetic Energy (J) | Bulge Height (mm) |
|---|---|---|---|---|---|
| 1-10 | 6 | 6 | 597 | 714 | 2.32 |
| 1-11 | 6 | 9 | 491 | 481 | 0.51 |
| 1-12 | 6 | 9 | 514 | 528 | 0.62 |
| 1-13 | 6 | 9 | 519 | 540 | 0.57 |
| 1-14 | 6 | 9 | 521 | 543 | 0.7 |
| 1-15 | 6 | 9 | 527 | 555 | 0.64 |
| 1-16 | 6 | 9 | 597 | 713 | 0.94 |
| 1-17 | 6 | 9 | 650 | 846 | 1.49 |
| 1-18 | 6 | 9 | 651 | 848 | 1.35 |
| 1-19 | 6 | 9 | 699 | 979 | 1.53 |
| 1-20 | 6 | 9 | 738 | 1089 | 1.75 |
| 1-21 | 6 | 9 | 739 | 1094 | 1.64 |
| 1-22 | 6 | 9 | 753 | 1135 | 2.07 |
| 1-23 | 6 | 9 | 800 | 1281 | 2.44 |

As shown in Table 3, the bulge height increased with the increase in the striking velocity and striking kinetic energy. The calculation formula of striking kinetic energy $E = 0.5mv^2$, where $m$ is fragment mass, and $v$ is fragment impact velocity. The bulge height is plotted in Figure 5. Although the bulge height increased with the increase in the velocity and kinetic energy of the fragment, the bulge height had a nonlinear relationship with the velocity of the fragment, whereas it changed linearly with the kinetic energy of the fragment. This provided key information for the derivation of the model presented below.

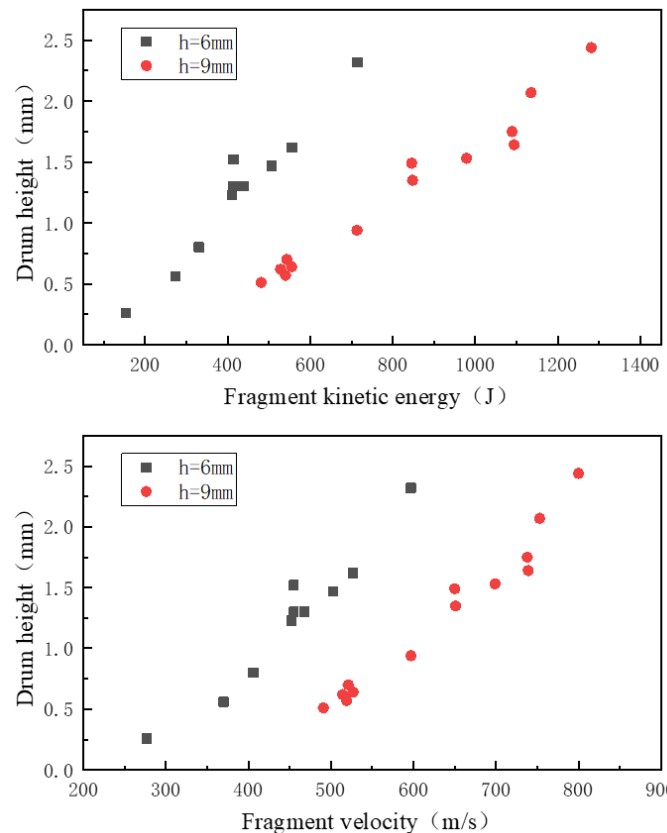

**Figure 5.** Variation of the bulge height with the fragment velocity and fragment kinetic energy at different target thickness.

## 3. Theoretical Analysis

The penetration of a tungsten fragment into a steel plate is a complex physical process, which is accompanied by the propagation of elastic–plastic waves, micro-deformation of the fragment, local deformation of the target plate, and stress transients [17]. Based on a few assumptions, we established a simplified model to describe the penetration of fragments into carbon-fiber target plates based on dimensional analysis.

In order to establish the formula, the impact process is simplified as follows:

(1)    Thermal effects are neglectable.
(2)    The fragment is a rigid body.
(3)    The fragment moves in a straight line.

Dimensional analysis is widely used by researchers as a tool to determine the relationship between various quantities in complex processes or physical phenomena. In the early twentieth century, Buckingham proposed that every law of physics could be expressed by several dimensionless quantities. Bridgman referred to this as the $\pi$ theorem, which became an important method in dimensional analysis [10].

The penetration process of a square fragment through a target plate is affected by many factors. The main factors and their dimensions in metric units are shown in Table 4. By analyzing each factor, a dimensionless equation was established.

**Table 4.** Main factors and their dimensions.

| Variable | Symbol | Dimension |
|---|---|---|
| Fragment length | $l$ | L |
| Fragment density | $\rho_f$ | $ML^{-3}$ |
| Fragment velocity | $v$ | $LT^{-1}$ |
| Target plate thickness | $h$ | L |
| Bulge height | $h_g$ | L |
| Density of the target plate | $\rho_t$ | $ML^{-3}$ |
| Strength of the target plate | $\sigma_t$ | $ML^{-1}T^{-2}$ |
| Strength of the fragment | $\sigma_f$ | $ML^{-1}T^{-2}$ |
| Fragment kinetic energy | $E$ | $ML^2T^{-2}$ |

The tests described above showed that the bulge height changed with the fragment energy. Based on dimensional analysis, the bulge height equation was established. The main factors involved in the penetration of the square fragment and their dimensions are shown in Table 4. There were nine characteristic parameters for the penetration process. When the mass of the fragment was known, the fragment energy and velocity could be calculated interchangeably. Therefore, the fragment energy was used to characterize the fragment velocity in this study. According to the above simplification, the fragment was regarded as a rigid body, and thus, the effect of the fragment strength on the penetration process was not considered. Hence, the characteristic parameters describing the process of square fragments penetrating the target plate were $l$, $\rho_f$, $h$, $h_g$, $\rho_t$, $\sigma_t$, and $E$.

The relationship between the bulge height of the target plate and the fragment energy is as follows:

$$1 = f\left(h, l, \rho_f, \rho_t, E, \sigma_t, h_g\right) \tag{1}$$

There are seven dimensional parameters in this equation. According to the $\pi$ theorem, Equation (1) can be replaced by $(n - k)$ dimensionless equations, where $k$ is the number of independent dimensions (here $k = 3$, so, $n - k = 4$). Based on past experience, three dimensionless parameters can be determined:

$$\pi_1 = l/h \tag{2}$$

$$\pi_2 = \rho_f/\rho_t \tag{3}$$

$$\pi_3 = h_g/h \tag{4}$$

In order to determine the last parameter containing the target plate strength $\sigma_t$, dimensional analysis was performed. It is assumed that the general form of the dimensionless parameter of the last term is

$$\pi_4 = h^{\lambda_1} \rho_t^{\lambda_2} \sigma_t^{\lambda_3} E^{\lambda_4} \tag{5}$$

Based on $[\pi_4] = M^0 L^0 T^0$,

$$\begin{aligned} \pi_4 &= (L)^{\lambda_1}(ML^{-3})^{\lambda_2}(ML^{-1}T^{-2})^{\lambda_3}(ML^2T^{-2})^{\lambda_4} \\ &= L^{\lambda_1 - 3\lambda_2 - \lambda_3 + 2\lambda_4} M^{\lambda_2 + \lambda_3 + \lambda_4} T^{\lambda_3 + \lambda_4} \end{aligned} \tag{6}$$

The resulting system of algebraic equations is

$$\begin{cases} \lambda_1 - 3\lambda_2 - \lambda_3 + 2\lambda_4 = 0 \\ \lambda_2 + \lambda_3 + \lambda_4 = 0 \\ \lambda_3 + \lambda_4 = 0 \end{cases} \tag{7}$$

and

$$\begin{cases} \lambda_4 = -\lambda_3 \\ \lambda_2 = 0 \\ \lambda_1 = 3\lambda_3 \end{cases} \tag{8}$$

In this study, let $\lambda_4 = 1$, and then $\lambda_3 = -1$ and $\lambda_1 = -3$. Thus,

$$\pi_4 = h^3 E / \sigma_t \tag{9}$$

Therefore, Equation (1) can be re-written as

$$1 = \left(\frac{l}{h}\right)^a \cdot \left(\frac{\rho_f}{\rho_t}\right)^b \left(\frac{h_g}{h}\right)^c \left(\frac{h^3 E}{\sigma_t}\right)^d \tag{10}$$

Through the above test, it is known that there is a linear relationship between the bulge height increment $\Delta h_g$ and the fragment energy increment $\Delta E$, and thus, c = −d = 1. Hence,

$$\Delta h_g = \left(\frac{l}{h}\right)^a \cdot \left(\frac{\rho_f}{\rho_t}\right)^b \frac{1}{h^2 \sigma_t} \cdot \Delta E \tag{11}$$

where $a$ and $b$ are undetermined constants.

According to the test results, the curve of the bulge height of the target plate with the fragment energy must pass through point $(E_0, 0)$, where $E_0$ is the minimum energy for the fragment to cause plastic deformation on the back of the target plate. The fragment energy equation can be written as follows:

$$h_g = \left(\frac{l}{h}\right)^a \cdot \left(\frac{\rho_t}{\rho_f}\right)^b \frac{1}{h^2 \sigma_t} \cdot (E - E_0) \tag{12}$$

Since $E_0$ is affected by the material properties of the target plate, the following equation can be established:

$$E_0 = \left(\frac{l}{h}\right)^x \cdot \left(\frac{\rho_f}{\rho_t}\right)^y h^3 \sigma_t \tag{13}$$

By integrating the equations, the following equations are obtained:

$$\begin{cases} h_g = A \cdot (E - E_0) \\ A = \left(\frac{l}{h}\right)^a \cdot \left(\frac{\rho_f}{\rho_t}\right)^b \frac{1}{h^2 \sigma_t} \\ E_0 = \left(\frac{l}{h}\right)^x \cdot \left(\frac{\rho_f}{\rho_t}\right)^y h^3 \sigma_t \end{cases} \tag{14}$$

As shown by Equation (14), under certain target conditions, $A$ and $E_0$ are constant values, and the penetration energy required increases with the increase in the fragment velocity. By fitting the data in Table 2 with Equation (14), the relationship between the fragment energy and the bulge height of the target plate was obtained, as shown in Figure 6. The fitting coefficient and fitting accuracy for different target plates are shown in Table 5.

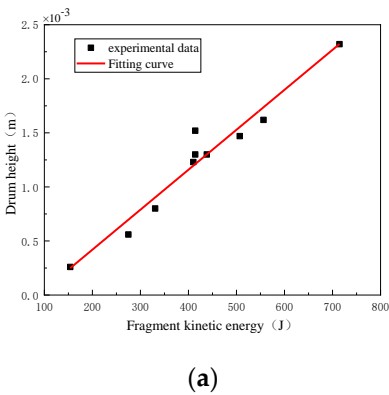

(**a**)

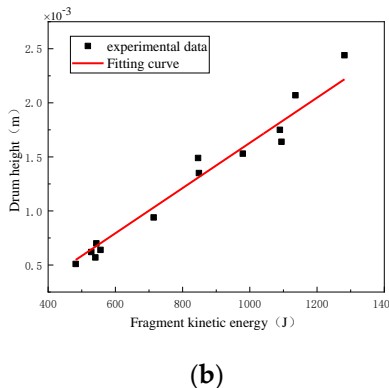

(**b**)

**Figure 6.** Variation of the bulge height with the fragment energy. (**a**) Condition 1 (fragment length: 6 mm, target plate thickness: 6 mm) (**b**) Condition 2 (fragment length: 6 mm, target plate thickness: 9 mm).

**Table 5.** Fitting coefficients and fitting accuracy under the same target system.

| Target System | Fitting Coefficient | | Fitting Accuracy |
|:---:|:---:|:---:|:---:|
| | $A$ | $E_0$ | |
| 1 | $3.7 \times 10^{-6}$ | 86.43 | 95.41 |
| 2 | $2.09 \times 10^{-6}$ | 220.31 | 96.35 |

Figure 6 shows that the bulge height of the target plate increased linearly with the increase in the fragment energy. By fitting the data, the parameters were obtained (Table 5). As the ratio of the side length of the fragment to the thickness of the target plate increased, the coefficient A increased, whereas the fitting coefficient $E_0$ decreased. In other words, the bulge height increased with the increase in the ratio of the side length of the fragment to the thickness of the target plate when the fragment energy remained the same, and the minimum energy for plastic deformation of the back of the plate was reduced accordingly. In addition, the fitting accuracies for the different target and projectile systems were 95% or better, indicating that Equation (14) is suitable for predicting the bulge height, and thus, it can be used in practical situations.

By obtaining the values of $A$ and $E_0$ under different conditions, the four parameters $a$, $b$, $x$, and $y$ were calculated, as shown in Table 6.

**Table 6.** Parameter calibration values.

| Parameter | $a$ | $b$ | $x$ | $y$ |
|:---:|:---:|:---:|:---:|:---:|
| Calibration value | −0.4911 | −2.5281 | 0.6924 | −0.8090 |

## 4. Validation

In order to validate the predictions of the bulge height of a gun steel target plate impacted by square tungsten alloy fragments, 6-mm square tungsten alloy fragments were used to carry out armor-piercing tests on gun steel target plates of different thicknesses, and a high-speed camera was used to measure the fragment velocities. The bulge height of

the target plate was calculated using Equation (14). The results are shown in Table 7. The calculation error is

$$\delta = \left| \frac{\text{Experimental values}_{h_g} - \text{Calculated value}_{h_g}}{\text{Experimental values}_{h_g}} \right| \times 100\% \tag{15}$$

Since Equation (14) was obtained by fitting the test results after dimensional analysis, it is suitable for the calculation of the bulge height of a gun steel target plate. The applicable parameter ranges are $v < 1400$ m/s, $0.42 < l/h < 1$. Some of the data in Table 7 were compared with the values calculated using the following equation [18]:

$$h_g = Avm^U \tag{16}$$

where $A$ and $U$ are the fitting coefficients, $A = 1.4 \times 10^{-3}$, and $U = 0.79$. $m$ is the mass of the fragment (kg), and $v$ is the striking velocity of the fragment (m/s).

**Table 7.** Fragment penetration tests.

| No. | Fragment Length (mm) | Target Plate Thickness (mm) | Striking Velocity (m/s) | Striking Kinetic Energy (J) | Bulge Height (mm) | | Relative Error |
|-----|-----|-----|-----|-----|-----|-----|-----|
| | | | | | Test Value | Calculated Value | |
| 2-1 | 6 | 14 | 742 | 1104.06 | 0.49 | 0.529 | 7.95 |
| 2-2 | 6 | 14 | 759 | 1153.52 | 0.56 | 0.582 | 3.92 |
| 2-3 | 6 | 14 | 883 | 1559.48 | 0.95 | 1.018 | 7.15 |
| 2-4 | 6 | 14 | 952 | 1813.61 | 1.16 | 1.290 | 11.20 |
| 2-5 | 6 | 14 | 1016 | 2065.27 | 1.57 | 1.561 | 0.57 |
| 2-6 | 6 | 14 | 1055 | 2228.85 | 1.7 | 1.736 | 2.11 |
| 2-7 | 6 | 14 | 1069 | 2288.88 | 1.73 | 1.800 | 4.04 |
| 2-8 | 6 | 14 | 1125 | 2534.84 | 2.12 | 2.065 | 2.59 |
| 2-9 | 6 | 14 | 1154 | 2663.38 | 2.62 | 2.203 | 15.91 |
| 2-10 | 6 | 14 | 1166 | 2720.88 | 2.27 | 2.264 | 0.26 |
| 2-11 | 6 | 14 | 1196 | 2860.38 | 2.54 | 2.414 | 4.96 |
| 2-12 | 6 | 14 | 1233 | 3040.44 | 2.54 | 2.607 | 2.63 |
| 2-13 | 6 | 14 | 1252 | 3133.44 | 3.17 | 2.707 | 14.60 |
| 2-14 | 6 | 11 | 500 | 500.94 | 0.21 | 0.232 | 10.47 |
| 2-15 | 6 | 11 | 535 | 571.67 | 0.34 | 0.342 | 0.58 |
| 2-16 | 6 | 11 | 622 | 775.15 | 0.58 | 0.656 | 13.10 |
| 2-17 | 6 | 11 | 696 | 970.20 | 0.83 | 0.957 | 15.30 |
| 2-18 | 6 | 11 | 732 | 1071.22 | 1.01 | 1.113 | 10.19 |
| 2-19 | 6 | 11 | 734 | 1077.47 | 1.05 | 1.123 | 6.95 |
| 2-20 | 6 | 11 | 769 | 1184.65 | 1.18 | 1.288 | 9.15 |
| 2-21 | 6 | 11 | 809 | 1307.69 | 1.41 | 1.478 | 4.82 |

As shown in Table 7, the average relative error between the bulge height calculated using Equation (14) and the test value was 7.07%, and the maximum error was 15.91%. In comparison, the relative error between the calculated value using Equation (16) and the test value was more than 90%. Therefore, the model established by the dimensional analysis in this study had a higher accuracy. Figure 7 shows that about 95% of the calculation points of the equation in this study were concentrated within 15%, and about 60% were within 10%. According to the above analysis, the model established in this study meets the requirements of engineering applications.

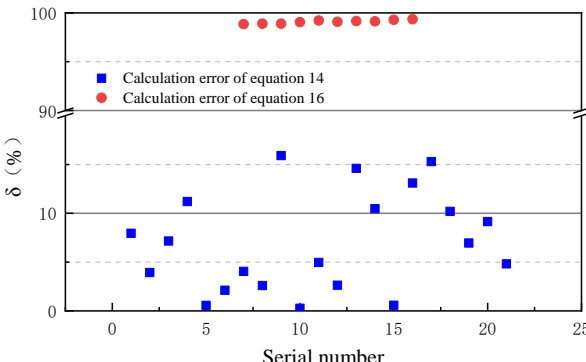

**Figure 7.** Error comparison.

Figure 8 shows that, as the thickness of the target plate changed, the bulge height increased linearly with the increase in the kinetic energy of the fragment. Under the same kinetic energy of the fragment, an increase in the thickness of the target plate led to a decrease in the bulge height. This was because when the fragment impacted the target plate, the target plate was compressed, and the energy absorbed by the homogeneous material was proportional to its volume. The greater the thickness of the target plate, the smaller the plastic deformation.

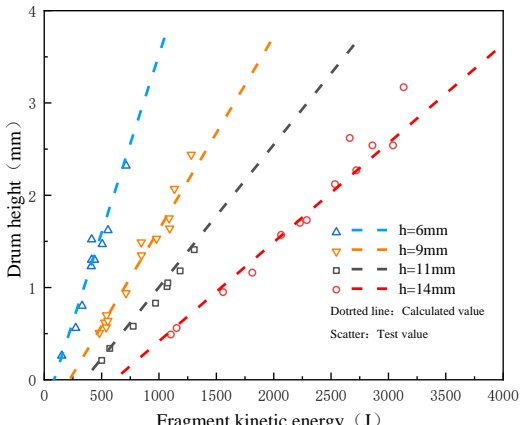

**Figure 8.** Variation of the bulge height with the fragment kinetic energy.

## 5. Conclusions

In this study, tungsten alloy fragments were used to impact gun steel target plates of different thicknesses, and quantitative data of the ballistic performance of the steel target plate were obtained. An equation of the bulge height of the steel target was established through dimensional analysis. The following conclusions were obtained.

For the same target system, when the tungsten alloy fragment did not penetrate the steel target plate, the bulge height of the target plate increased with the increase in the kinetic energy of the fragment, and their relationship was linear. When the fragment penetrated the target plate with the same kinetic energy, an increase in the thickness of the target plate led to a decrease in the bulge height. The relationship between the bulge height and the kinetic energy was obtained through dimensional analysis. The calculated values were in good agreement with the test values, and the error was within 16%. Thus, the equation meets the requirements of engineering applications. Based on the damage modes of the target plate, cratering, compression, shearing, and tensile failure were the main failure modes of the steel target plate when the tungsten alloy fragment did not penetrate the plate. With increasing impact energy, the pit depth of the target plate became significantly larger than the bulge height of the target plate.

**Author Contributions:** Methodology, D.H. and Y.Z.; Validation, N.J., Z.L. and Y.J.; Writing—original draft, D.H.; Writing—review & editing, D.H., W.L., K.Z. and Y.J. All authors have read and agreed to the published version of the manuscript.

**Funding:** This research received no external funding.

**Conflicts of Interest:** The authors declare no conflict of interest.

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
