# Peer review of "Study on Bulge State of PCrNi3MoV Target Plates Penetrated by Square Fragments"

_applsci, doi:10.3390/app12146987_

Round 1

Reviewer 1 Report

Referee report on “Study on Bulge State of PCrNi3MoV Target Plates Penetrated 2 by Square Fragments’’ by Dou Hong, Wen-bin Li, Yu Zheng, Ning Jiang and Ke-bin Zhang

The authors considered the penetration characteristics of tungsten alloy fragments on PCrNi3MoV target plates. Specifically, studied the penetration of square tungsten alloy fragments on gun steel target plates of different thicknesses using a ballistic gun launching device. Based on the ballistic test results, the authors were able to establish the relationship between the bulge height of the target plate and the kinetic energy of the fragments and analyzed the failure mode and process of the tungsten alloy fragments penetrating the steel target. Dimensional analysis was used to ascertain the variation of the bulge height of the target plate with the kinetic energy of the fragments.

This paper is well-written, scientifically sound, and I did not have any problem following the logic of the paper. The mathematics seems sound, and the potential real-world application of the predictive model make it an interesting study worth publication.

The authors should improve upon the consistency of the list of references (e.g., issue, volume, page numbers etc).

Author Response

Answer: Thank you for your suggestion. I have improved the consistency of the reference list and marked the modified part in red in the reference list. But some of the older literatures can not collect more complete information.

References

  1. Hafizoglu H, Durlu N, Konokman H E. Effects of sintering temperature and Ni/Fe ratio on ballistic performance of tungsten heavy alloy fragments[J]. International Journal of Refractory Metals Hard Materials, 2019, 81: 155-166.
  2. Zhu F L, Chen Y, Zhu G L. Numerical simulation study on penetration performance of depleted Uranium (DU) alloy fragments[J]. Defence Technology, 2021, 17(1): 50-55.
  3. Dey S, Børvik T, Hopperstad O S, et al. On the influence of fracture criterion in projectile impact of steel plates[J]. Computational Materials Science, 2006, 38(1): 176-191.
  4. Huang C Q, Zhu H S. Establishment of the formula for the ultimate penetration velocity of spherical fragments to target [J] JJournal of Projectiles,Rockets,Missiles and Guidance, 1993, (2): 58-61.
  5. Zhang Q M, Huang F L .Limit velocity of fragment penetrating target and equivalent target [J] Acta Armamentarii, 1996, (1): 21-25.
  6. Zhao W H, Zhu X S. Model establishment and calculation of spherical fragment penetrating finite thick target [J] Journal of Ballistics, 1997, (3): 20-23.
  7. Cheng C, Du Z, Chen X. Oblique penetration of tungsten alloy rod to finite-thickness metal plate[J]. Latin American Journal of Solids and Structures, 2019, 16 (09).
  8. Zhao G. Simplified model for oblique penetration of long-rod into plates of finite thickness[J]. Acta Armamentarii, 1987.
  9. Recht R F, Ipson T W. Ballistic Perforation Dynamics[J]. Journal of Applied Mechanics, 1963, 30:384.
  10. Schäfer F, Herrwerth M, J.Hiermaier S, et al. Shape effects in hypervelocity impact on semi-infinite metallic targets[J]. International Journal of Impact Engineering, 2001, 26(1): 699-711.
  11. Martineau R L, Prime M B, Duffey T. Penetration of HSLA-100 Steel with Tungsten Carbide Spheres at Striking Velocities Between 0.8 and 2.5 km/s[J]. International Journal of Impact Engineering, 2004, 30(5): 505-520.
  12. Xu Y X, Wang S S, Bo X F, et al. Armor-piercing ultimate of tungsten alloy spherical fragment against low-carbon steel [J]
    Journal of Vibration and Shock, 2011, (05): 192-195.
  13. Xu J X, Yuan J J, Deng Y F. Research on Impact Performance of Layer-number of 2024 Aerospace Aluminum Alloy Immobilized Target Against Bird Strike [J] Journal of System Simulation, 2018, 30 (010): 3914-3922.
  14. Mo G L, Wu Z L, Liu K. Analytical model of projectile penetrating ceramic-UHMWPE composite armor [J] Journal of Nanjing University of science and technology, 2013, (5): 5.
  15. Keil T, Taheriniya S, Bruder E, et al. Effects of solutes on thermal stability, microstructure and mechanical properties in CrMnFeCoNi based alloys after high pressure torsion[J]. Acta Materialia, 2022, 227(24): 117689.
  16. Pei S X, Wu X M, Sun T, et al. Study on penetration of tungsten ball into multilayer target [J] Journal of Projectiles,Rockets,Missiles and Guidance, 1993, (2): 23-30.
  17. Hong D, Li W B, Zheng Y, et al. Experimental research on tungsten alloy spherical fragments penetrating into carbon fiber target plate[J]. Latin American Journal of Solids and Structures, 2021, 18(5):14
  18. Shen X J, Zhang P X, Sun T, et al. Study on criteria of gun barrel failure caused by fragment impact of explosive warhead [J] Projectiles,Rockets,Missiles and Guidance, 2006, 26 (1):84-86.

Reviewer 2 Report

The article describes penetration tests of steel plates with square-section tungsten elements. The authors, based on the concept of dimensional analysis, found a relationship between the height of the bulge in the plate and the kinetic energy of the tungsten elements, and verify it at other plate thicknesses not previously tested.

The article is written in a clear and orderly manner. The research steps taken have their justification. The research scheme presented in the article is indeed identical to the other work of the authors quoted under number [16], but it concerns spherical tungsten fragments, which seems justified taking into account practice. Therefore, I would like an explanation from the authors what was the motivation for considering the pieces with a square cross-section, not the rather common spherical ones. Are there any new features in this article compared to the one presented earlier? It is quoted in the context of a dimensional analysis but I would suggest a comparison taking into account the novelty elements in the present article.

The photos in Figures 1 and 3 are not clear and too small. It would be good to improve them.

After supplementing these issues, I believe that the article can be published.

Reviewer 3 Report

The topic of this article is one of the important and useful topics in the industry. Also written in a good scientific way.

Author Response

(1) There is a syntax error on line 115 of the original text. The calculation formula of Striking kinetic energy in Table 3 needs to be listed. The table names in Table 5 need to be modified.

Answer: Thank you for your suggestion. The syntax error in line 115 of the original text and the improper wording in Figure 5 in line 154 have been modified. The formula of striking kinetic energy in Table 3 has been described later, "the calculation formula of striking kinetic energy E=0.5mv2, where m is fragment mass, and V is fragment impact velocity." Changes in the text have been marked in red.

Reviewer 4 Report

In this paper authors utilized tungsten fragments to invenstigate the impact on the steel target plates. 

This is a solid and interesting work. However, considering the high impact of the Applied Sciences, I want to suggest certain and important improvements. 

1- It would be very useful if authors can provide a detailed micrograph such as scanning electron microscopy (SEM) picture of the before and after the target penetration. It will be extremely useful to observe the morphological changes. 

2- Figure 4 and 5 should be combined into the same figure. As it is now, figure 4 actually does not represent much. Also please use higher quality pictures. Using a better camera, I suggest taking new pictures and changing the current ones with the new high quality.  

3- In the theoretical analysis, authors neglected the thermal effects. But, thermal expansion during the target penetration should have an effect on the depth. Why did you choose to neglect? Can you please explain with relevant literature if exists. 

4- Please check the grammar. ı have encountered some minor mistakes. 

Round 2

Reviewer 4 Report

I had suggested SEM pictures of the before and after the bullet damage. It would be a nice addition to see morphological changes. 

Apart from that, Authors provided an answer letter to a reviewer who had concerns on repetitive study as he has noticed a very similar paper from the authors. I agree with him/her as applied sciences is a high quality journal. Hence, I would suggest publising if those concerns are addressed and repetivive work is removed or changed. 

Author Response

This manuscript is a resubmission of an earlier submission. The following is a list of the peer review reports and author responses from that submission.